# Determinants of early initiation of breastfeeding following birth in West Africa: A multilevel analysis using data from multi-country national health surveys

**Bewuketu Terefe** [1]*, **Tadele Biresaw Belachew**[2], **Desale Bihonegn Asmamaw**[3], **Gizachew Tadesse Wassie**[4], **Abebaw Gedef Azene**[4], **Habitu Birhan Eshetu**[5], **Kindie Fentahun Muchie**[4], **Getasew Mulat Bantie**[6], **Kassawmar Angaw Bogale**[4], **Wubshet Debebe Negash**[2]

1 Department of Community Health Nursing, School of Nursing, College of Medicine and Health Sciences, University of Gondar, Gondar, Ethiopia, 2 Department of Health Systems and Policy, Institute of Public Health, College of Medicine and Health Sciences, University of Gondar, Gondar, Ethiopia, 3 Department of Reproductive Health, Institute of Public Health, College of Medicine and Health Sciences, University of Gondar, Gondar, Ethiopia, 4 Department of Epidemiology and Biostatistics, School of Public Health, College of Medicine and Health Sciences, Bahir Dar University, Bahir Dar, Ethiopia, 5 Department of Health Promotion and Health Behavior, Institute of Public Health, College of Medicine and Health Sciences, University of Gondar, Gondar, Ethiopia, 6 Amhara National Regional State Public Health Institute, Amhara Region, Ethiopia

* woldeabwomariam@gmail.com

## Abstract

### Background

Early initiation of breastfeeding (EIBF), within the first hour of birth, is crucial for promoting exclusive breastfeeding and establishing optimal nursing practices. However, global EIBF rates remain low, with even lower rates observed in Africa. Despite existing research gaps, this study aims to determine the prevalence of EIBF and identify maternal and child-related factors associated with its practice in West Africa.

### Methods

This study utilized West African Demographic and Health Survey (DHS) data from 13 countries, including 146,964 children's records. To assess model fit, likelihood test and deviance were used. Similarly, intraclass correlation coefficient, median odds ratio, and proportional change in variance were employed for random effect. A multilevel logistic regression model was used to identify individual- and community-level factors influencing EIBF due to the hierarchical nature of the data. Variables with p-values ≤0.2 in the binary model and <0.05 in the final analysis were considered significantly associated with EIBF.

### Results

The pooled prevalence of EIBF in West African nations was 50.60% (95% CI; 50.34–50.85%). The highest prevalence rate was observed in Serra Leone (75.33%) and the

**Data Availability Statement:** All relevant data are in the manuscript and its Supporting Information files.

**Funding:** The author(s) received no specific funding for this work.

**Competing interests:** The authors have declared that no competing interests exist.

lowest prevalence was found in Senegal (33.94%). In the multilevel multiple logistic regression model, maternal education (AOR = 1.10, 95% CI, 1.03,1.16), marital status AOR = 1.07, 95% CI, 1.01,1.13), birth weight (AOR = 0.91, CI 0.86,0.96), birth orders (AOR = 1.09, CI 1.03,1.16), and (AOR = 1.11, CI 1.03,1.19), place of residence (AOR = 1.14, CI 1.07,1.21), and mode of delivery type (AOR = 0.26, CI 0.24,0.29) were significantly correlated with EIBF in West Africa.

## Conclusions

The incidence of EIBF in West Africa was found to be low. The study emphasizes the need for targeted behavioral change communication programs to address timely breastfeeding initiation, specifically targeting mothers and child characteristics. Factors such as education, delivery mode, marital status, birth weight, birth order, and place of residence were significantly associated with EIBF. Special attention should be given to improving EIBF rates among women undergoing caesarean sections, infants with low birth weight, and primiparous mothers, along with structural improvements in the healthcare sector in West Africa.

## Introduction

Early initiation of breastfeeding (EIBF) can be defined as the number of newborns who breastfed within one hour of birth during a specified reference period, with the inclusion of expressed breast milk defined as the percentage of the total number of live births in the same period [1]. In general, breast milk contributes to the health and well-being of newborns and mothers, and EIBF within an hour after birth is an irreplaceable measure in averting neonatal and infant morbidity and mortality by regulating the body temperature in skin-to-skin contact, stimulating breast milk production, enhancing the uterine workforce, and reducing bleeding [2, 3]. EIBF within an hour benefit not only the neonates, but also the mother, who will be able to profitably initiate lactation, sustain exclusive breastfeeding over the long term, reduce the incidence of postpartum hemorrhage, minimize problems breastfeeding, and maintain favorable breastfeeding intention and behavior [4–6]. Even though the World Health Organization (WHO) has established a strong recommendation that breastfeeding should begin within the first hour of birth with a continuation of exclusive breastfeeding until six months [7], several developing countries could not have acted accordingly and achieved the intended outcome and plan of sustainable development goal of indicator number three [8, 9]; however, if optimal breastfeeding is implemented ultimately, these countries could save 22% and 13% death of neonates and under-five [10–12] respectively,; on the other hand, optimum culture of breastfeeding could save about 800,000 under five and a rate reduction of 55% to 87% globally by preventing death caused by several infections such as diarrhea, sepsis, and pneumonia [13, 14]. Another modeling study also revealed that community workers will have a probability of saving 4.9 million death [15]; however, half of the newborns did not receive the first one hour breastfeeding according to the 2017 WHO report [16].

In sub-Saharan Africa (SSA), the EIBF has faced multidimensional challenges from the commitment of government stakeholders, health care providers, health facility structures, and accessibility, as well as from the perspective of the community and mothers [17]. Several studies have shown that EIBF coverage of EIBF was shockingly low and varies across countries. The prevalence of EIBF in SSA is 52.8%, with a significant variation from 17% in Guinea to

95% in Malawi [18, 19]. Another similar study in SSA also showed that 58.2% prevalence of EIBF ranging from 24% in Chad and 86% in Burundi [20].

Numerous studies have been conducted to identify potential hindering factors related to why mothers and health care providers did not start EIBF, and factors such as pain, discomfort and physical vulnerabilities, cultural beliefs about EIBF, colostrum as a dirty food, furnishment of prelacteal feeding, and preferences for certain food were among the top listed categories of why mothers did not intend to start EIBF [21–23]. On the other hand, other determinant factors such as, education, occupation, age, wealth index, breast disease, knowledge, place of delivery, antenatal care follow up, mode of delivery, birth order of the neonate, residency, distance to health facility and pregnancy intention were among the factors that are affected EIBF greatly [9, 24–29].

West Africa nations are among the top listed countries with poor progress both in EIBF and continues breastfeeding practices, for example the 2017 UNICEF, reported depicted that EIBF was 52% in Gambia [30], however the most recent demographic and health survey of the Gambia said that EIBF was reduced from 52% in 2013 to 36% in 2019–20 [31]. A study conducted on the skin-to-skin approach among mothers and newborns was low (35.7%) [32], and another study in the same place also recommended that countries have been severely affected by traditional beliefs and that EIBF is still low with high neonatal deaths [33].

The existing body of research on EIBF in West Africa is limited in several ways. Firstly, the number of studies conducted on this topic is scarce, indicating a lack of comprehensive understanding of the factors influencing EIBF in the region. Moreover, these studies have often focused on only a small portion of the population, limiting the generalizability of their findings to the broader West African context. Furthermore, previous studies have primarily examined individual-level factors associated with EIBF, such as maternal characteristics or infant health indicators. While these individual-level factors are undoubtedly important, there is a need to explore and understand the broader implementation factors that influence EIBF practices in West Africa. These implementation factors could include community-level dynamics, healthcare systems, cultural practices, and policy environments that may impact the initiation of breastfeeding within the region. To address these research gaps, the aim of our study was to assess the possible implementation factors influencing EIBF in West Africa. By taking a broader perspective and considering factors beyond the individual level, we sought to provide a more comprehensive understanding of the determinants and barriers to EIBF in the region. The scientific findings of this study will help the West African government and other stakeholders, policymakers and program implementers who have a lot of work to improve the reduced EIBF by developing a timely policy.

## Methods and materials

### Study area, and period

The Westernmost part of Africa is West Africa or Western Africa. The 16 countries of Benin, Burkina Faso, Cape Verde, The Gambia, Ghana, Guinea, Guinea-Bissau, Ivory Coast, Liberia, Mali, Mauritania, Niger, Nigeria, Senegal, Sierra Leone, and Togo are included in the United Nations definition of Western Africa, along with Saint Helena, Ascension, and Tristan da Cunha (a United Kingdom Overseas Territory) [34, 35]. According to estimates, West Africa may have a total population of 419 million people by 2021 and 381,981,000 by 2017, with 189,672,000 females and 192,309,000 males [36]. One of the fastest-expanding regions on the African continent is the region in question, both economically and demographically.

We combined data from the children's files of the 13 West African countries that participate in the Demographic and Health Surveys (DHS) program and for which data have been

published in the previous five years: Benin, Burkina Faso, Ghana, Gambia, Guinea, Liberia, Mali, Mauritania, Niger, Nigeria, Senegal, Sierra Leone, and Togo. However, EIBF was recorded in f13 countries only. Every five years, surveys of low- and middle-income countries are conducted nationwide for DHS. The survey we used was conducted in West African nations between 2012 and 2022.

## Study populations

Mothers between the ages of 15 and 49 years were interviewed. Based on our inclusion criteria, we selected a sample of women from among those who were interviewed. The study included last-born children (those born within the two years prior to the survey and those under 24 months old), as well as newborns who were clothed in their mothers' bare skin. A total weighted sample of 146,964 mothers who had ever breastfed and who had living children under the age of two was used in this investigation, using the children's dataset.

## Data sources and sampling procedures

Recent nationally representative DHS data collected from 13 West African countries served as the data source for this study. EIBF data were not collected from 3 out of 16 nations; therefore, we included 13 nations only in this study. Every five years, low- and middle-income nations participate in DHS surveys, which are consistently conducted using pretested, validated, and standardized techniques. Multi-country analysis was made possible by using the same standard process for sampling, questionnaires, data collection, and coding.

The DHS uses a stratified two-stage sampling approach to guarantee national representativeness. Clusters/enumeration areas (EAs) that encompass the entire nation were randomly chosen from the sample frame in the first step (i.e., typically created from the most recently available national census). The second stage involves systematically selecting houses from each cluster or EA and conducting interviews with selected households' target demographics (women between the ages of 15 and 49 and men between the ages of 15 and 64, respectively). Women between the ages of 15 and 49 who gave birth within the previous five years and who either had or did not have an EIBF exposure for their most recent child were included in this study. Pooled data analysis yielded a total weighted sample size of 146,964, and unweighted of 147,400 participants with sample sizes ranging from 34,193 in Nigeria to 5,263 in Liberia (Table 1, and Fig 1).

## Data collection, quality control

In the DHS, a pre-test was conducted before data collection, a debriefing session with pre-test fieldworkers was held, and changes to the questionnaires were made as necessary. The DHS guidance provided additional details regarding the data-collection process. The details can be accessed from the Guide to DHS statistics.

## Study variables

**Outcome variable.** In children under the age of 24 months, we examined the influence of social and personal factors on the start of breastfeeding immediately before delivery. We classified the timely or immediate commencement of breastfeeding in these children using WHO and DHS guidelines. The study outcome variable was the timing of breastfeeding. It originated from the query, "How long after birth did you first put (NAME) to the breast?". The answers were immediately available on hours and days [37, 38]. The replies were then divided into two

**Table 1. Countries, survey year, and samples of demographic and health surveys included in the analysis for 13 West African countries.**

| Country | Survey year | Unweight frequency | Weighted frequency |
|---|---|---|---|
| Burkina Faso | 2022 | 14,965 | 15,292 |
| Benin | 2017/18 | 13,589 | 13,643 |
| Ghana | 2014 | 5,884 | 5,695 |
| The Gambia | 2019/20 | 8,362 | 7,653 |
| Guinea | 2018 | 7,951 | 7,885 |
| Liberia | 2019/20 | 5,704 | 5,263 |
| Mali | 2018 | 9,940 | 10,304 |
| Mauritania | 2019/21 | 11,628 | 11,685 |
| Nigeria | 2018 | 33,924 | 34,193 |
| Niger | 2012 | 12,450 | 13,255 |
| Serra Leone | 2019 | 9,899 | 9,771 |
| Senegal | 2019 | 6,125 | 5,618 |
| Togo | 2013/14 | 6,979 | 6,706 |
| Total | | 147,400 | 146,964 |

categories: timely initiation of breastfeeding = 1 (if women said they started immediately after giving birth or within the first hour) and late initiation = 0 (if they did not) [37, 38].

**Independent variables.** Individual level (Level 1) and community level (Level 2) variables were used to categorize the independent variables. Individual-level factors included maternal age, marital status, maternal education, employment status, wealth quantiles, media exposure, antenatal follow-up, size of the child at birth, birth order, place of delivery, type of delivery, and twin status. In contrast, community variables included variables that were directly collected (such as residence and region) and variables obtained by aggregating individual variables into their respective communities (such as community media exposure, community poverty, community women's education, community ANC coverage, and community health institution delivery utilization rate). Based on the data distributions, the mean and median values of the percentage of women in each category of a given variable were used to calculate the aggregate. We divided the aggregate values of a cluster into groups based on the median values because the aggregate values of each variable did not follow a normal distribution curve.

## Operational definitions

Independent variables were classified after reviewing similar literatures as follows (Table 2)

## Definitions of community level variables

Place of residence and area were variables at the non-aggregate community level. The location of habitation is listed as either urban or rural. The children's home province was used to describe the location. To comprehend the neighborhood effect on the implementation of EIBF, another group of community-level variables was established by aggregating from an individual level and utilizing averaging techniques. neighborhood wealth, neighborhood media exposure, neighborhood education for women, and neighborhood birthplace.

**Community female education.** This is the aggregate value of women's educational levels based on the average proportions of educational levels in the community. It was defined as low if the ratio of women with secondary education and above in the community was below the median, and high if the value was higher than the median. The median value was 20.69%

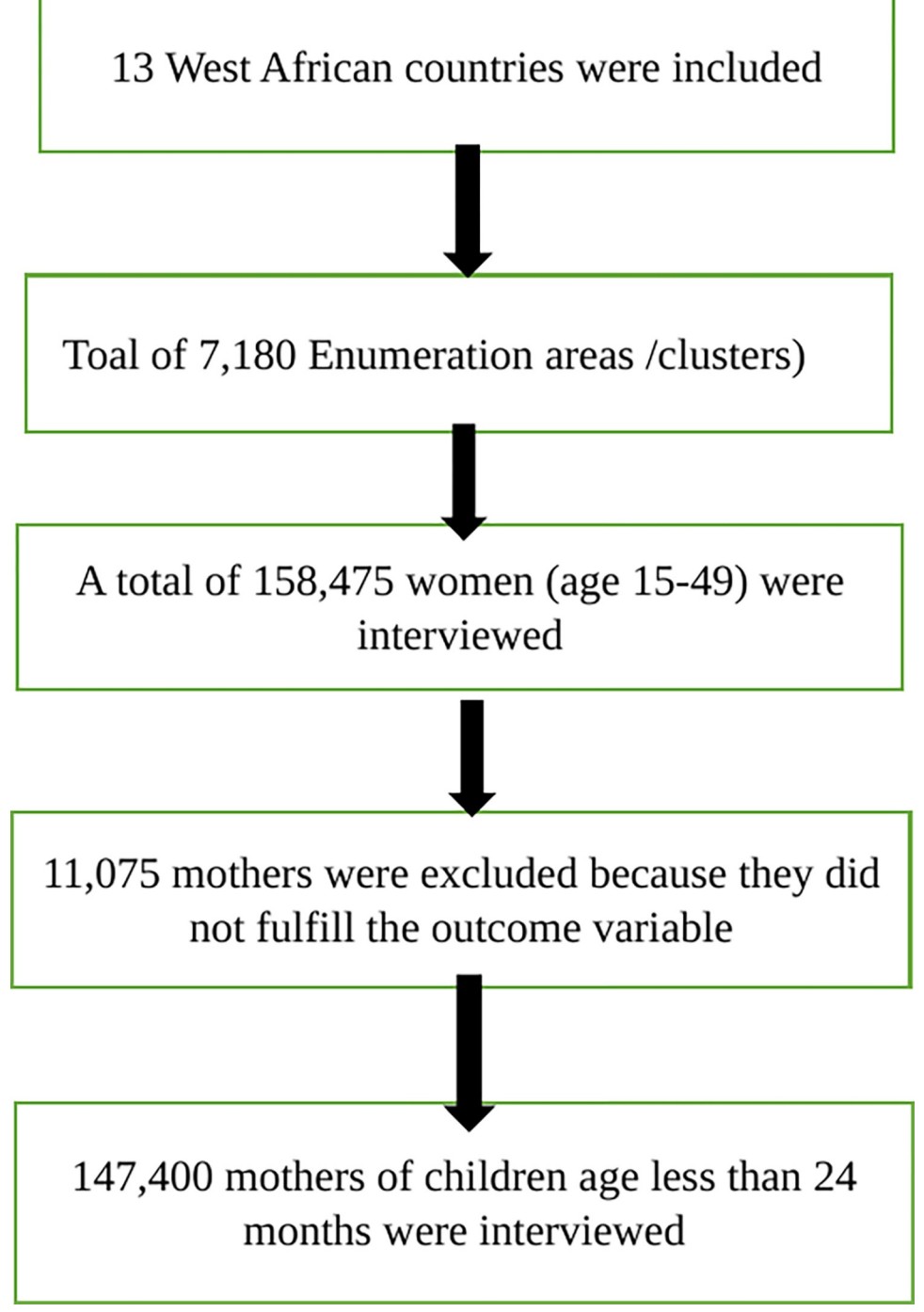

**Fig 1. Schematic representation of the sampling procedures in the study of determinants of d early initiation breastfeeding in West African countries.**

**Community media exposure.** This variable was derived from individual responses to reading magazines, listening to radio, and watching television. It was defined as low if the proportion of women exposed to media in the community is low and high if the community media exposure level was equal to or greater than the median value (35.72%)

**Table 2. Shows operational classification of independent variables (Table 2).**

| | |
|---|---|
| Antenatal care visit | Antenatal care follow-up was also expressed in number in the DHS data set, and we have recategorized it at least one antenatal care visit (Yes), if not "No" |
| Place of delivery | Place of delivery was classified as at home and health facility delivery. |
| Maternal age | Maternal age was classified as 15–24, 25–34 and 35–49 years old |
| Maternal education | Maternal education was classified as no formal education, primary, and secondary and high educational level |
| Wealth index | Wealth index was classified as poor, middle and rich respectively |
| Caesarean section | Caesarean section was given as yes or no option in the DHS and we have used it as it is |
| Marital status | Marital status was considered as not married and married |
| Maternal occupation | Maternal occupation was classified as have employed and have not been employed |
| Birth order | Order of birth was given numerically in the original data set, and as usual, we have reorganized it as first, from 2–3, and ≥4 birth orders |
| Size of the child at birth | The size of the child at birth was classified in the GDHS data set as very small, smaller than average, average, larger than average, and very large depending on the mother's perception of the size of the child at birth. We reclassified as small-size babies, average-size babies, and large-size babies, falling into each of these categories |
| Twin status | We classified children as either singletons or twins if there were more than one child |
| Sex of child | We categorized the sex of the child as either female or male when classifying them |
| Community level wealth | We categorized community-level women wealth as either low or high based on the proportion value. |
| Community level women education | We categorized community-level women education as either low or high based on the proportion value. |
| Community ANC coverage | We categorized community-level women antenatal care coverage as either low or high based on the proportion value. |
| Community level health facility delivery coverage | We categorized community level women health facility delivery coverage as either low or high based on the proportion value. |
| Community level media exposure | Similarly, we have categorized community level women mass media exposure coverage as either low or high based on the proportion value. |

**Community health facility delivery utilization rate.** This variable is also derived from the individual values for the place of delivery utilization. It was defined as low if the median proportion of women who attended health facility delivery in the community was lower than the median value (82.54%), and high if the proportion was equal to or greater than the computed median value.

**Community wealth status.** This variable is derived from an individual household's wealth index using the same procedure. It was defined as high if the proportion of women from the two low-wealth quintiles and the highest medium, richer, and richest quantiles in a given community. For this proportion, summary statistics were produced, and the calculated median value (49.09%) was used to classify the fraction as "high" or "low."

## Data processing, procedure, and analysis

Data were extracted from Kids Records (KR) files using statistical software, including STATA version 17 and Microsoft Excel 19. Additional coding and transformation were performed. Prior to analysis, the sample's sampling weight and normalization were carried out to ensure that it was representative of the various regions and respondents' dwellings. Weighted samples were used in the analysis to correct for unequal selection and non-response probabilities in the

original survey. The demographic and health survey (DHS) data of West African countries have a hierarchical structure because the data were gathered using multi-stage stratified cluster sampling techniques. Single-level logistic regression is not recommended because typical multiple regression methods treat the units of analysis as separate observations. Failure to recognize hierarchical structures leads to an overestimation of the standard errors of regression coefficients, which increases the statistical significance. According to this perspective, a complex statistical model that considers the hierarchical structure of the data is required to draw significant conclusions. Therefore, a multivariable multilevel binary logistic regression model was used to evaluate the fixed and random effects of the anemia-related factors. Four models were created with the knowledge that communities (clusters) will have different intercepts, but fixed coefficients. The first model, which is sometimes referred to as an empty or null model because it was fitted without any explanatory factors, was used to compare the differences in EIBF status between the clusters of the sample. This model, which was created to break down the variance that occurred between communities, was used to compare the variation in EIBF status among clusters in the data. Model one, however, was used to assess how individual-level variables influenced the variation in EIBF status after controlling for them. Model two was used to examine whether community-level variables were accountable for between-cluster heterogeneity in childhood EIBF after taking community-level factors into account. By merging individual and community-level data. Using the null model as a baseline, we calculated the percentage of community features that might explain the observed variability in EIBF. Furthermore, this model served as a litmus test to determine whether multilevel or conventional logistic regression should be used, supporting the use of a multilevel statistical framework. It was assessed using the Log-Likelihood Ratio test (LLR), Median Odds Ratio (MOR), intraclass correlation coefficient (ICC), and Proportional Change of Variance (PCV). Moreover, the model comparison was made using model deviance, a model with the lowest deviance selected for reporting and interpreting results.

## Null model

For individual $i$ in community j, the model can be represented as [39, 40]:

$$Y_{ij} + \Upsilon_{00} + u_{0j} + \varepsilon_{ij} \ldots\ldots\ldots\ldots\ldots\ldots\ldots\ldots\ldots\ldots\ldots\ldots\ldots\text{null model}$$

Where: $Y_{ij}$ is the EIBF status of $i^{th}$ child in the $j^{th}$ cluster, $\mu_{00}$ = is the intercept; that is the probability of having EIBF exposure in the absence of explanatory variables, $\mu_{0j}$ = community-level effect; $\varepsilon_{ij}$ error at the individual level.

Mixed model: This model was derived by simultaneously mixing individual- and community-level factors [41].

$$Y_{ij} + \Upsilon_{00} + \Upsilon_{k0} X_{kij} + \Upsilon_{0p} z_{pj} + u_{0j} + \varepsilon_{ij} \ldots\ldots$$

where $\gamma_{k0}$ is the regression coefficient of the individual-level variables $X_k$, and $\gamma_{0p}$ is the regression coefficient of the community-level variable $Zp$. $X_k$ and $Zp$ were individual and community-level explanatory variables respectively. Subscripts $i$ and $j$ represent the individual level and cluster number, respectively.

Intraclass correlation coefficient (ICC): a measure of within-cluster homogeneity and the proportion of variance due to between-cluster differences.

ICC = $\delta^2_{\mu0} \div \delta^2_{\mu0} {}_{+\pi}{}^2/3$, where: $\delta^2_{\mu0}$ = between cluster (community) variances and $\pi^2/3$ = within the cluster (community) variance. The value of $\pi^2/3$ in the case of standard logistic distribution is 3.30

According to the null model, there were significant differences in the EIBF status between clusters (p = 0.001; 2u0 = 0.46). The ICC was 12.3% (95% CI: .10.82,14.01), indicating that unobserved factors at the community or individual level or differences between communities accounted for 12.3% of the overall variability in EIBF risk. This suggests that a multilevel logistic regression model is preferable to a single-level logistic regression model for obtaining reliable results [39]. From 12.3% in the null model to 11.9%, 11.9%, and 11.6% in models one, two, and three, respectively, the variation attributable to the clustering effect was reduced (Table 6).

Proportion Change in Variance (PCV): Calculated using the null model to determine the contribution of each factor to the variation in childhood EIBF.

$$\text{PCV} = (\delta^2_{\mu0-}\delta^2_{\mu1})100 \div \delta^2_{\mu0.}$$

where $\delta^2_{\mu0}$ is the difference between the community variances in the null model and $\delta^2_{\mu1}$ is the difference between the community variances in the following models [17]. Model three's PCV, which was 32%, was higher than that of the null model. This demonstrates that the simultaneous effects of both individual- and community-level factors found in the model were responsible for 43% of the variance in the anemia status of the children. Regarding the variance metrics (random effects), area variance (AV) and its 95% confidence interval (CI), the intraclass correlation coefficient (ICC), median odds ratio (MOR), and proportional change in variance were some of the indicators used to evaluate the measures of variation. Similarly, to test for model fit, the Akaike information criterion (AIC), whose lower value indicates a better model fit, the lower AIC was used to evaluate each model's goodness-of-fit. To evaluate multicollinearity, the variance inflation factor (VIF) and tolerance were used. No variables had multicollinearity issues (all VIF <10 and tolerance > 0.1). The overall mean VIF was 1.99 (Table 6).

The association between the independent variable and EIBF was first investigated using binary logistic regression analysis. Second, multilevel fixed effect modelling analysis methodologies were used to examine the components connected to the EIBF (p-value 0.2). We employed a manual backward stepwise elimination method by ranking the covariates in order of importance, because this process for selecting variables allowed us to consider the impact of each variable. Both unadjusted and adjusted odds ratios (AOR) are displayed with 95% confidence intervals (CI). Finally, a p value < 0.05 was used to demonstrate statistical significance.

**Ethical considerations and data set access.** The study was conducted after obtaining a permission letter from www.dhsprogram.com on an online request to access DHS data after reviewing the submitted brief descriptions of the survey to the DHS measure program office. The datasets were treated with the utmost confidence. This study was done based on secondary data from recent West African countries DHS. Ethical clearance was not applicable, and there was no direct contact with women. Issues related to informed consent, confidentiality, anonymity, and privacy of the study participants are already done ethically by the DHS office. We did not manipulate and apply the microdata other than in this study. There was no patient or public involvement in this study.

## Results

### Maternal related sociodemographic characteristics of the study participants

In this study, 146,964 participants were enrolled in West African countries. Approximately half of them 72,508 (49.34%) of the study participants were found from 25–34 years of

reproductive age. Regarding marital status, the majority of the mothers 127,489 (86.75%) were married. With respect to place of delivery, 91,750 (62.43%) and mode of delivery type 141,122 (96.10%) higher proportion of mothers gave birth at health institutions and not delivered by cesarean section, respectively. In addition, the majority 135,327 (92.08%0 of mother had followed ANC at least once during their pregnancy period. More than half 86,011 (58.53%) did not enroll in formal school. Approximately 64,519 (43.90%) and 90,371 (61.49%) mothers showed poor wealth index status and were currently employed, respectively (Table 3).

### Community level characteristics

More than half (96,966; 65.98%) of the study participants were from rural residential areas. Regarding to another community level variables of wealth index 74,902 (50.97%), community women educational status 74,421 (51.32%), media exposure 74,193 (50.48%), and health facility coverage 72,234 (50.64%) have come from high level of wealthy, educated, media exposure, and low facility-based delivery communities, respectively (Table 4)

### Children related characteristics

Nearly half of the 74,823 (50.91%) and 74,877 (50.95%) children were male and had an average birth weight, respectively. Most children 67,285 (45.78%) had 4th and more orders of birth. In contrast, 141,087 (96.00%) children were born single (Table 5).

### Pooled prevalence of EIBF in West Africa

The pooled prevalence of EIBF in West African countries was 50.60%, with a 95% confidence interval of (50.34–50.85%). Serra Leone had the highest prevalence of EIBF (75.33%), whereas Senegal had the lowest prevalence (33.94%). Furthermore, Niger (52.85%), Benin (53.52%0, Ghana (55.10%), Mauritania (55.83%), Togo (60.71%), Mali (63.37%), and Liberia (64.47%),

**Table 3. Maternal related sociodemographic characteristics in West African countries (n = 146,964).**

| Variables | Category | Weighted Frequency | Percentage |
|---|---|---|---|
| Maternal age in years | 15–24 | 37,506 | 25.52 |
| | 25–34 | 72,508 | 49.34 |
| | 35–49 | 36,951 | 25.14 |
| Marital status | Unmarried | 19,475 | 13.25 |
| | Married | 127,489 | 86.75 |
| Educational status | Not educated | 86,011 | 58.53 |
| | Primary | 26,420 | 17.98 |
| | Secondary and higher | 34,533 | 23.50 |
| Wealth index | Poor | 64,519 | 43.90 |
| | Middle | 30,056 | 20.45 |
| | Rich | 52,389 | 35.65 |
| Occupational status | Not employed | 56,593 | 38.51 |
| | Employed | 90,371 | 61.49 |
| At least one ANC follow up | No | 11,637 | 7.92 |
| | Yes | 135,327 | 92.08 |
| Place of delivery | Home | 55,214 | 37.57 |
| | Health facility | 91,750 | 62.43 |
| Delivered by caesarean section | No | 141,122 | 96.10 |
| | Yes | 5,842 | 3.90 |

**Table 4. Community level variables characteristics among reproductive age mothers in West Africa countries (n = 146,964).**

| Variables | Category | Frequency | Percentage |
|---|---|---|---|
| Types of residence | Urban | 49,998 | 34.02 |
| | Rural | 96,966 | 65.98 |
| Community level wealth | Low | 72,062 | 49.03 |
| | High | 74,902 | 50.97 |
| Community level women education | Low | 71,543 | 48.68 |
| | High | 74,421 | 51.32 |
| Community ANC coverage | Low | 70,234 | 47.79 |
| | High | 76,730 | 52.21 |
| Community level health facility delivery coverage | Low | 72,234 | 50.64 |
| | High | 74,730 | 49.36 |
| Community level media exposure | Low | 72,771 | 49.52 |
| | High | 74,193 | 50.48 |

were countries which scored above the pooled prevalence results of EIBF among West African countries respectively (Fig 2).

## Random effect and factors associated with EIBF in West African countries

This study used multilevel logistic regression to fit the model. Hence, four fitted models, including the null model, model one, model two, and model three were used to show the fixed and random effects. In the null model, there was a significant variance in the probability of children in West Africa being exposed to EIBF (community-level variance = 0.56, p <0.001). As implied by the intra-cluster correlation coefficient (ICC) in the empty model, regional differences would account for 12.3% of the variation in children's EIBF. Furthermore, the median odds ratio (MOR) was 1.91(1.82,2.00), this can be interpreted as when children go from low to high EIBF prevalence area, the likelihood of exposing for EIBF was 1.91 times higher. The PCV of this study was 26.79%, which indicates that both country- and individual-level variables explained 26.79% of the national variation observed in an empty model. Determinates such as maternal education, wealth index, marital status, birth weight, birth order, delivery by cesarean section, and types of residence were statistically significant in the multilevel multivariable logistic regression model in West Africa.

The odds of being exposed to EIBF among children increased by 90% (AOR = 1.10, 95% CI, 1.03,1.16) among women who had no formal education enrollment as compared to

**Table 5. Children related demographic and health survey characteristics in West African countries(n = 146,964).**

| Variables | Category | Weighted Frequency | Percentage |
|---|---|---|---|
| Sex of the child | Male | 74,823 | 50.91 |
| | Female | 72,141 | 49.09 |
| Birth order | First | 29,657 | 20.18 |
| | 2$^{nd}$ and 3$^{rd}$ | 50,022 | 34.04 |
| | 4$^{th}$ and above | 67,285 | 45.78 |
| Size of the child at birth | Small | 26,071 | 17.94 |
| | Average | 74,877 | 50.95 |
| | Large | 46,016 | 31.31 |
| Twin status | Single | 141,087 | 96.00 |
| | Multiple | 5,877 | 4.00 |

**Table 6. Individual and community-level factors associated with EIBF among reproductive-age women in West Africa (n = 146,964).**

| Independent variables | Null model | Model I | Model II | Model III |
|---|---|---|---|---|
| | | AOR [95% CI] | AOR [95% CI] | AOR [95% CI] |
| Maternal age | | | | |
| 15–24 | | 1 | | 1 |
| 25–34 | | 0.98(0.92,1.03) | | 1.03(0.97,1.09) |
| 35–49 | | 0.98(0.92,1.06) | | 0.99(0.91,1.06) |
| Education | | | | |
| No formal education | | 1.02(0.96,1.07) | | 1.10(1.03,1.16) * |
| Primary | | 0.92(0.87,0.98) | | 0.99(0.93,1.06) |
| Secondary and higher | | 1 | | 1 |
| Wealth index | | | | |
| Poor | | 1.16(1.09,1.23) | | 1.06(0.99,1.13) |
| Middle | | 1.02(0.96,1.09) | | 0.95(0.89,1.02) |
| Rich | | 1 | | 1 |
| Marital status | | | | |
| Unmarried | | 1 | | 1 |
| Married | | 0.88(0.84,0.93) | | 1.07(1.01,1.13) * |
| Sex of the child | | | | |
| Male | | 1 | | 1 |
| Female | | 0.97(0.93,1.01) | | 0.96(0.93,1.01) |
| Child is twin | | | | |
| No | | 1 | | 1 |
| Yes | | 1.05(0.95,1.16) | | 1.05(0.95,1.16) |
| Birth weight | | | | |
| Larger | | 1.04(0.98,1.08) | | 1.02(0.97,1.07) |
| Average | | 1 | | 1 |
| Smaller | | 0.90(0.85,0.94) | | 0.91(0.86,0.96) * |
| Birth order | | | | |
| First | | 1 | | 1 |
| 2[rd] and 3[th] | | 1.12(1.06,1.19) | | 1.09(1.03,1.16) * |
| 4[th] and above | | 1.12(1.04,1.19) | | 1.11(1.03,1.19) * |
| Place of delivery | | | | |
| Home | | 1 | | 1 |
| Health facility | | 0.97(0.92,1.02) | | 0.99(0.94,1.05) |
| Media exposure | | | | |
| No | | 1 | | 1 |
| Yes | | 0.79(0.76,0.83) | | 1.03(98,1.08) |
| Delivered by CS | | | | |
| No | | 1 | | 1 |
| Yes | | 0.28(0.25,0.31) | | 0.26(0.24,0.29) * |
| Type of residence | | | | |
| Urban | | | 1 | 1 |
| Rural | | | 1.25(1.19,1.31) | 1.14(1.07,1.21) * |
| Community media exposure | | | | |
| Low | | | 1 | 1 |
| High | | | 0.99(0.90,1.11) | 0.99(0.89,1.11) |
| Community-women education | | | | |
| Low | | | 1 | 1 |

(*Continued*)

**Table 6.** (Continued)

| Independent variables | Null model | Model I | Model II | Model III |
|---|---|---|---|---|
| | | AOR [95% CI] | AOR [95% CI] | AOR [95% CI] |
| High | | | 0.96(0.87,1.07) | 1.00(0.90,1.12) |
| Community-level wealth | | | | |
| Low | | | 1 | 1 |
| High | | | 0.88(0.78,.99) | 0.92(0.82,1.04) |
| Community health facility delivery coverage | | | | |
| Low | | | 1 | 1 |
| High | | | 0.97(0.86,1.09) | 0.98(0.88,1.11) |
| **Random parameters and model comparison** | **Null model** | **Model I** | **Model II** | **Model III** |
| Community-level variance | 0.56(0.50,0.64) | 0.44(.38,.51) | 0.44(0.38.0.52) | 0.41(0.35,0.48) |
| ICC (%) | 12.3 | 11.9 | 11.9 | 11.6 |
| MOR (95% CI) | 1.91(1.82,2.00) | 1.89(1.80,1.98) | 1.89(1.80,1.98) | 1.88(1.79,1.96) |
| PCV | Reference | 21.43 | 21.43 | 26.79 |
| Log-likelihood (LLR) | -30832.70 | -30297.70 | -29555.90 | -29140.1 |
| DIC (-2LLR) | 61,665.4 | 60,595.4 | 59,111.8 | 58,280.2 |
| AIC | 61669.4 | 60633.4 | 59133.8 | 58336.2 |

*Indicates significance at p-value <0.05 variables in the regression

women with secondary and higher levels. EIBF imitating increased by 1.07 times among children whose mothers married when compared with children whose mothers unmarried (AOR = 1.07, 95% CI, 1.01,1.13). Children whose birth weight was lower showed a lower likelihood of feeding EIBF within one hour after birth (AOR = 0.91, CI 0.86,0.96) compared to average birth-weighted children.

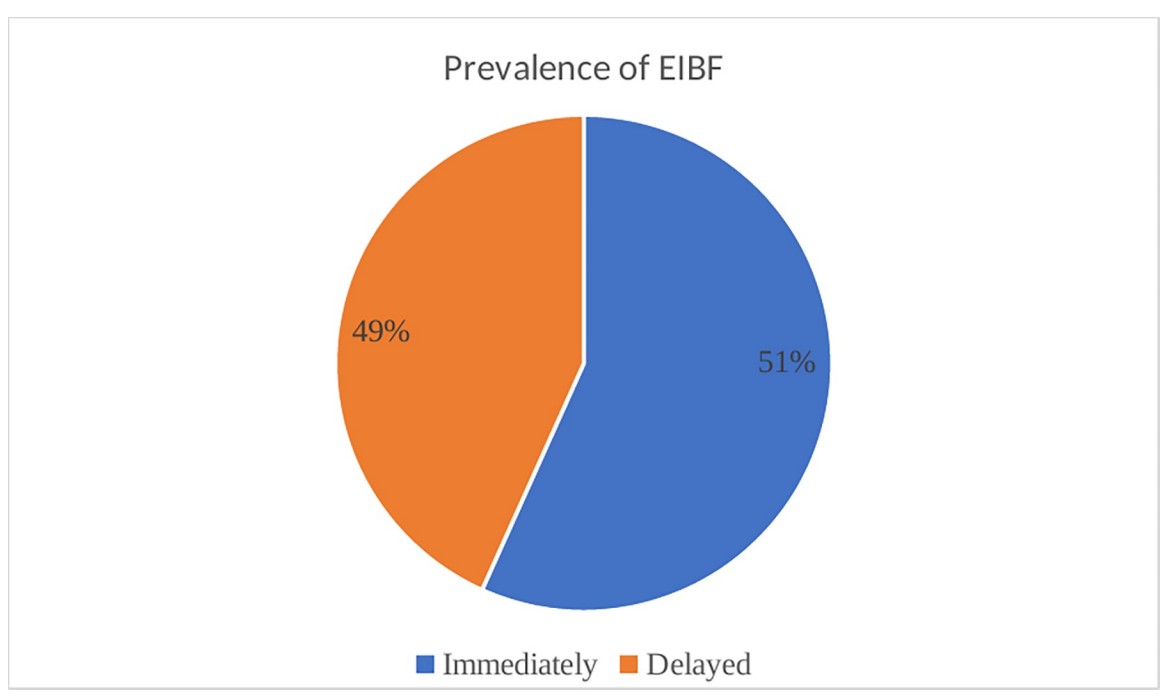

**Fig 2. Pooled prevalence of EIBF in West Africa countries.**

Regarding birth order, children with 2nd and 3rd birth and above order of birth showed a higher likelihood of being exposed to EIBF within one hour following birth as compared to first birth order children by the odds of (AOR = 1.09, CI 1.03,1.16), and (AOR = 1.11, CI 1.03,1.19) respectively. Mothers who have come from rural areas and those who have given their birth by cesarean section have shown a more likelihood and a less likelihood chances to expose their children to EIBF within an hour following to their birth by the odds of (AOR = 1.14, CI 1.07,1.21), and (AOR = 0.26, CI 0.24,0.29) respectively (Table 6).

## Discussion

The main goal of this study was to use the latest Demographic and Health datasets to examine the pooled prevalence and determinants of early initiation of breastfeeding practices among mothers in West Africa who have children under the age of five from 2012 to 2022. In this study, we discovered that the combined prevalence of EIBF in Sub-Saharan Africa was 57%, which is less than the recommendations of the WHO and UNICEF [42, 43]. The findings showed that, among individuals living in West Africa, timely breastfeeding initiation remains a priority for public health. In light of the aforementioned, the current investigation found that particular mother and child characteristics are linked to timely breastfeeding initiation.

According to this study, there is a negative correlation between a child's birth weight and the likelihood that breastfeeding will begin promptly. This correlation was particularly strong for mothers of smaller-than-average children. It is conceivable that some newborns who were underweight may be maturing through physiological processes related to the formation of the "nutritive sucking pathway" [44, 45]. Therefore, it may not be possible to place a newborn within an hour of birth when health professionals use basic and advanced neonatal care procedures to save their lives. Limiting their ability to nurse within the first hour of life so requires that they have effective coordination of the suction-deglutition respiratory cycle and the breast-seeking response [45, 46]. A premature baby's poor sucking ability, swallowing challenges, and poor coordination, as well as the need for admission to an NICU due to conditions such as respiratory distress, jaundice, meconium aspiration, and low birth weight, could be the possible causes of the delay in EIBF.

In contrast to a prior study [47–49], the current investigation found a negative correlation between a mother's education level and timely breastfeeding initiation. Our observation was unexpected because, according to some, education encourages women to be open to health information, which sharpens their behavior and reorients them to choose positive health behaviors, such as adopting healthy infant feeding habits including timely commencement of breastfeeding [50], however a study conducted by Francis Appiah and his colleagues has revealed similar association between EIBF and level education [45]. It is also crucial to highlight that uneducated women are more willing to help and less likely to dispute a doctor's request. Additionally, this woman can have cultural views on breastfeeding [51, 52]. Owing to the cross-sectional nature of the current study, we advise against attempting to explain the causes of this association. Therefore, it will be helpful to conduct further research to clarify the connection between these maternal factors and the prompt commencement of breastfeeding.

The timing of breastfeeding initiation positively correlated with marital status. Married women have shown an odd of 1.07 times to start EIBF compared to unmarried mothers. Possible explanations for this include the possibility that married mothers benefit from their husbands' positive support, encouragement, and care more than unmarried mothers do [53, 54]. Mothers who are married may also have a higher likelihood of desired and planned pregnancies [55]. This then enables them to start fetal interventions for the mother, the child, and other sociocultural factors.

Timing of breastfeeding initiation was negatively correlated with caesarean delivery. Yisma et al. found that caesarean sections were linked to a 46% lower incidence of timely breastfeeding initiation among 33 countries in sub-Saharan Africa in their meta-analysis of the effects of caesarean sections on breastfeeding indicators [56]. In addition to this, research by Birhan T. Y. et al and Appiah F. et al revealed comparable results in regards to the mode of delivery [45, 48]. It is argued that mothers experience significant discomfort following a cesarean section, which causes the practice of nursing to be delayed [46]. The effects of anesthesia may delay the start of nursing or cause respiratory distress in babies born by cesarean section, according to other research [47].

For mothers who lived in a rural rather than an urban area, the odds of breastfeeding being started on schedule increased by 1.14 times. Rural women in Saudi Arabia's Al-Hassa area were 4.2 times more likely to start breastfeeding within an hour than urban ones [57]. Various studies have shown that having good knowledge and living in urban areas does not necessarily mean the only factor to begin EIBF [52, 58]. From this vantage point, we can guarantee that, as long as we encourage rural mothers through healthcare delivery and campaigns, it is conceivable to boost the EIBF practice rate even more than in metropolitan areas. As they want to know everything possible about their children, these mothers are more likely to trust and listen to what health professionals have to say. This study also demonstrated a relationship between birth order and the timely commencement of breastfeeding, with women at parity 2-3rd and higher having increased probabilities of doing so. With each birth, mothers of higher parity probably improve their breastfeeding knowledge and techniques [45, 47], which benefits their nursing habits as a result of their first child's lived experience.

## Strength and limitations

The uniqueness and power of this study comes from the fact that it used the most recent datasets from five nationally representative surveys to examine both mother and child characteristics related to the timely initiation of breastfeeding in West Africa. Additionally, the study's robust findings are the result of the probability strategy used to choose survey respondents and a suitable analytical process. Furthermore, the adoption of a two-stage sampling procedure ensured that the outcomes were not tainted by selection bias. However, caution should be exercised when interpreting our findings. Additionally, the up to 5-year gap between birth and the interview can cause mothers to give false information about how long they breastfed their children due to recollection bias. In this study, social desirability bias cannot be avoided because the surveyed women may have provided information about the practices of prompt commencement of breastfeeding in an effort to project a favorable breastfeeding image among those who are aware of its benefits. Once more, the odds should be interpreted carefully because, as we acknowledge, the increased odds ratios seen in relation to some of the variables could be the result of the large sample size utilized in this investigation. Furthermore, the addition of some sociocultural variables, such as colonial history, which were absent from the dataset could have raised the percentage variance in the models. moreover, we acknowledge that our study's design prevented us from examining the causes of discrepancies, including mother and child characteristics and the timing of breastfeeding initiation. Finally, we acknowledge that the absence of time as an independent variable in our study may limit the interpretation of the results. Readers should exercise caution when interpreting the findings in light of this limitation.

## Conclusions

In West Africa, less than two out of every three mothers begin breastfeeding within an hour of giving birth. Therefore, this study found that EIBF practices in West Africa are still below the

benchmark set by WHO, UNICEF, and national policies. The level of education attained by the mother, delivery method, marital status, birth order, birth weight, type of place of residence, and country of residence were the maternal characteristics identified to have an impact and found to be statistically significant to the early commencement of breastfeeding. In particular, for Gambian mothers, for whom timely commencement of breastfeeding is still difficult, policies should be prioritized to improve the timely initiation of breastfeeding. mothers, especially those who undergo a cesarean section, and birth children that are smaller than average should receive enough supportive care in addition to advice and counseling in order to address breastfeeding inequities that are specific to this group of mothers. To reverse and close the gaps in the timely initiation of breastfeeding stratified by mother and child characteristics, behavior modification communication programs focused on this behavior should be conducted in West Africa. It is crucial to keep in mind that additional research that incorporates overlooked elements and spatiotemporal epidemiological studies will also play a role.

## Supporting information

**S1 Data.**
(DTA)

## Acknowledgments

We would like to acknowledge the DHS program for providing permission for this study following research ethics.

## Author Contributions

**Conceptualization:** Bewuketu Terefe.

**Data curation:** Bewuketu Terefe, Tadele Biresaw Belachew, Gizachew Tadesse Wassie, Habitu Birhan Eshetu, Wubshet Debebe Negash.

**Formal analysis:** Bewuketu Terefe, Tadele Biresaw Belachew, Abebaw Gedef Azene, Kassawmar Angaw Bogale.

**Funding acquisition:** Abebaw Gedef Azene.

**Investigation:** Desale Bihonegn Asmamaw, Abebaw Gedef Azene.

**Methodology:** Bewuketu Terefe, Desale Bihonegn Asmamaw, Gizachew Tadesse Wassie, Kindie Fentahun Muchie, Getasew Mulat Bantie, Wubshet Debebe Negash.

**Project administration:** Abebaw Gedef Azene, Getasew Mulat Bantie, Kassawmar Angaw Bogale, Wubshet Debebe Negash.

**Resources:** Gizachew Tadesse Wassie, Abebaw Gedef Azene, Habitu Birhan Eshetu, Kindie Fentahun Muchie, Getasew Mulat Bantie, Kassawmar Angaw Bogale.

**Software:** Bewuketu Terefe, Habitu Birhan Eshetu, Getasew Mulat Bantie, Kassawmar Angaw Bogale.

**Supervision:** Desale Bihonegn Asmamaw, Kindie Fentahun Muchie.

**Validation:** Bewuketu Terefe, Abebaw Gedef Azene, Habitu Birhan Eshetu, Kindie Fentahun Muchie, Kassawmar Angaw Bogale, Wubshet Debebe Negash.

**Visualization:** Gizachew Tadesse Wassie, Getasew Mulat Bantie, Kassawmar Angaw Bogale, Wubshet Debebe Negash.

**Writing – original draft:** Bewuketu Terefe, Wubshet Debebe Negash.

**Writing – review & editing:** Desale Bihonegn Asmamaw, Gizachew Tadesse Wassie, Habitu Birhan Eshetu, Kindie Fentahun Muchie, Getasew Mulat Bantie, Kassawmar Angaw Bogale, Wubshet Debebe Negash.

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
