## [Decision Letter · Decision Letter 0]

8 Mar 2024

PONE-D-23-39493Maternal and child related determinants of early initiation of breastfeeding following to birth in West Africa: further analysis using multilevel model from multi country national health surveyPLOS ONE

Dear Dr. Terefe,

Thank you for submitting your manuscript to PLOS ONE. After careful consideration, we feel that it has merit but does not fully meet PLOS ONE’s publication criteria as it currently stands. Therefore, we invite you to submit a revised version of the manuscript that addresses the points raised during the review process.

We look forward to receiving your revised manuscript.

Kind regards,

Anthony Mwinilanaa Tampah-Naah

Academic Editor

PLOS ONE

Journal Requirements:

3. We note that your Data Availability Statement is currently as follows: "All relevant data are within the manuscript and its Supporting Information files."

5. Please include a separate caption for your figure in your manuscript.

Reviewers' comments:

Reviewer's Responses to Questions

**Comments to the Author**

1. Is the manuscript technically sound, and do the data support the conclusions?

Reviewer #1: Yes

Reviewer #2: Yes

2. Has the statistical analysis been performed appropriately and rigorously? 

Reviewer #1: Yes

Reviewer #2: Yes

3. Have the authors made all data underlying the findings in their manuscript fully available?

Reviewer #1: Yes

Reviewer #2: Yes

4. Is the manuscript presented in an intelligible fashion and written in standard English?

Reviewer #1: Yes

Reviewer #2: Yes

5. Review Comments to the Author

Reviewer #1: This study was done before the COVID-19 pandemic. It would be very beneficial if there were also data referring to post-pandemic era. Kindly see comments in the uploaded word document. thank you so much.

Reviewer #2: General comment

The title is interesting and essential for the reduction of neonatal and children morbidity and mortality in the world. It is suitable to the journal of plose one.

Specific comments

1. Title: it is better to rearrange the title as “Determinants of early initiation of breastfeeding following to birth in West Africa: further analysis using multilevel model from multi country national health survey”. Why you said maternal and child related determinants???

2. Introduction: on the introduction part of your study various determinant factors of early initiation of breast feeding were explored by various international reports and studies. What is the significance of this study particularly using DHS?

3. Methods: well expressed and written

4. Result : from table 2 under the variable educational status there is poor ,middle and rich categories. What is this?

5. Discussion: well written

6. PLOS authors have the option to publish the peer review history of their article (what does this mean?). If published, this will include your full peer review and any attached files.

Reviewer #1: No

Reviewer #2: No

---

## [Author Response · Author response to Decision Letter 0]

10 Mar 2024

Response to the Review Comments 

Dear editor and reviewers

First for all the authors would like to thank the editor(s) and reviewers for your consideration, precious time, thoughtful comments and constructive suggestions, which help to improve the quality of this manuscript. We have responded to each critique/ comment and believe that the manuscript is much improved with the changes we made as suggested by the editor and reviewers. The corresponding changes and refinements made in the revised manuscript are summarized in our response below.

1. Is the manuscript technically sound, and do the data support the conclusions?

Reviewer #1: Yes

Reviewer #2: Yes

Response: Dear reviewers, thank you very much for your feedback and cross validation for our work. Please see our detail response below. Thank you ________________________________________2. Has the statistical analysis been performed appropriately and rigorously?

Reviewer #1: Yes

Reviewer #2: Yes

Response: Dear reviewers, thank you very much for your feedback and cross validation for our work. Please see our detail response below. Thank you ________________________________________3. Have the authors made all data underlying the findings in their manuscript fully available?

Reviewer #1: Yes

Reviewer #2: Yes

Response: Dear reviewers, thank you very much for your feedback and cross validation for our work. Please see our detail response below. Thank you ________________________________________4. Is the manuscript presented in an intelligible fashion and written in standard English?

Reviewer #1: Yes

Reviewer #2: Yes

Response: Dear reviewers, thank you very much for your feedback and cross validation for our work. Please see our detail response below. Thank you 

Reviewer #1:

 This study was done before the COVID-19 pandemic. It would be very beneficial if there were also data referring to post-pandemic era. Kindly see comments in the uploaded word document. thank you so much.

Response: Dear Reviewer, thank you for your feedback on our study. We appreciate your suggestion to include data referring to the post-pandemic era. We understand the value of examining the potential impact of the COVID-19 pandemic on our study's subject matter. However, we would like to clarify that our study is based on the data from the DHS (Demographic and Health Surveys), as a result, we do not have access to data specifically related to the post-pandemic era within the scope of our study. While we acknowledge the importance of investigating the potential effects of the pandemic on the topic of our study, it would require a separate research effort and data collection specifically focused on the post-pandemic period. We appreciate your thorough review and the comments provided in the uploaded word document, and we will address them accordingly to improve the quality and clarity of our manuscript. Thank you once again for your valuable feedback.

1. The objectives and rationale of the study were clearly stated. However, the abstract word count is higher than expected. I believe it is too much. The abstract should be reduced to an acceptable word count.

Response: Dear Reviewer, thank you for your valuable feedback regarding our study. We appreciate your acknowledgment that the objectives and rationale of the study were clearly stated. We understand your concern regarding the word count of the abstract exceeding expectations. We carefully reviewed the abstract and make necessary revisions to reduce its word count while ensuring that all essential information is retained. Our aim is to present a comprehensive overview of the study's key findings, methodology, and implications within the acceptable word count limits. We focused on condensing the abstract without compromising the clarity and coherence of the content. By doing so, we believe we can effectively communicate the core aspects of the study to readers while adhering to the required word count.

2. Authors should accurately state the purpose of the intraclass correlation coefficient (ICC), median odds ratio (MOR), and proportional change in variance (PCV) as they are random components and are not used to evaluate model fitness.

Response: Dear reviewer, thank you for your feedback on our study. We appreciate your comments regarding the accurate statement of the purpose of certain statistical measures, namely the intraclass correlation coefficient (ICC), median odds ratio (MOR), and proportional change in variance (PCV). We apologize for any confusion or misinterpretation in our manuscript. We understand that these measures are random components and are not typically used to evaluate model fitness. Instead, they serve different purposes in statistical analysis. To rectify this issue, we revised our manuscript to provide a clear and accurate explanation of the intended purposes of these measures. We ensured that the correct interpretation and use of these statistical measures are presented, aligning with the established literature and best practices.

3. In the abstract section, the authors mention that they have used the kids recorded data set, but they also report that they have used combined data from the women's files. This needs clarification.

 Response: Dear Reviewer, thank you for your feedback on our study. We appreciate your observation regarding the mention of using both the kids recorded (KR) dataset and combined data from the women's files in the abstract section. To clarify, we would like to confirm that we have only utilized the kids recorded (KR) dataset for our analysis. In our revised manuscript, we made sure to accurately state that our study exclusively relies on the kids recorded (KR) dataset. We also provided a more precise and clear description of the dataset used in our methods (data source subsection) to avoid any confusion or ambiguity.

4. The authors should involve the sample size determination procedure via table, graph, or diagram that was used and should be referenced as well as clearly stated.

Response: Dear reviewer, thank you for your feedback on our study. We appreciate your suggestion to involve the sample size determination procedure in a more explicit manner, using a table, graph, or diagram, and referencing it appropriately. We agreed that providing clear information about the sample size determination procedure is essential for transparency and reproducibility. In our revised manuscript, we incorporated a dedicated section or subsection that explicitly outlines the sample size determination procedure. This visual representation will enable readers to better understand the considerations and methods employed to arrive at the chosen sample size. Thank you for your valuable input. 

5. Authors should state both the dependent and independent variables in tabular form for easy comprehension by readers who will eventually have access to them in the long run.

Response: Dear reviewer, thank you for your feedback on our study. We appreciate your suggestion to present the dependent and independent variables in tabular form to facilitate easy comprehension for readers, both in the immediate context and in the long run. We agreed that tabulating the variables would provide a clear and concise overview of the variables included in our analysis. In our revised manuscript, we incorporated a table that presents the dependent and independent variables used in the study. Thank you for your valuable suggestion.

6. Weighting DHS data for a single country is different from weighting DHS data for multicounty analysis. Even though the authors used DHS data from 13 countries, they used sampling weight, which is used for a single-country DHS data weighting technique. Instead, the authors have to use other weighting techniques, for instance, differential weighing...

Response: Dear reviewer, thank you for your feedback on our study. We appreciate your comment regarding the weighting technique used in our analysis of the DHS data from multiple countries. We understand your suggestion to consider alternative weighting techniques, such as differential weighing, for multi-country analyses. However, it is important to note that the use of sampling weights provided by the DHS program is a widely accepted and recommended practice for analyzing DHS data across multiple countries. The DHS program provides explicit guidelines and recommendations on the appropriate use of sampling weights for multi-country analyses. These weights are designed to account for the complex survey design and ensure that the results are representative at the national and subnational levels. 

While other weighting techniques, such as differential weighing, may have their merits in certain contexts, the use of sampling weights provided by the DHS program is considered appropriate and standard for analyzing data from multiple countries within the DHS framework. To address your concern, we will clarify in our revised manuscript that we have followed the recommended practice of using sampling weights provided by the DHS program for our multi-country analysis. Thank you for bringing this to our attention.

7. DHS data may have different formats, standards, definitions, or classifications that make them difficult to compare, link, or aggregate across countries. How do authors handle the above challenges during multicounty analysis?

Response: Dear Reviewer, thank you for your feedback on our study. We appreciate your comment regarding the potential challenges in comparing, linking, or aggregating DHS data across multiple countries. We understand your concern and would like to address it. We acknowledge that the DHS data may have variations in formats, standards, definitions, or classifications across countries. However, it is important to note that the DHS program has implemented measures to ensure consistency and comparability of the data across different countries. 

The DHS program follows standardized protocols and guidelines for data collection, which include consistent survey instruments, questionnaires, and procedures. These protocols aim to minimize differences in data collection processes across countries, thereby facilitating comparability. In our study, we have employed rigorous procedures to handle the challenges associated with multi-country analysis. We have carefully reviewed and standardized the data formats, definitions, and classifications to ensure consistency and comparability across countries.

Additionally, we have appended the country-specific data files based on the enumeration area numbers (from the smallest to the largest EAs), following a logical and systematic approach. This approach enables us to organize the data in a consistent and coherent manner, facilitating accurate analysis and interpretation.

Furthermore, we have referenced the DHS program's guidelines and documentation to ensure the correct interpretation and application of variables and indicators, including the definition of exclusive breastfeeding.

In our revised manuscript, we will provide explicit details on the steps taken to address the challenges of comparing, linking, and aggregating DHS data across multiple countries. This will help readers understand our approach and the measures we have taken to ensure the accuracy and reliability of our analysis. Thank you for bringing this to our attention.

8. It is good that the authors appended the West African countries DHS data for secondary analysis, but the big concern is how authors address the effect of the variation of time to combine data sets for analysis.

Response: Dear reviewer, thank you for your feedback on our study. We appreciate your comment regarding the potential effect of combining data sets from West African countries that may have variations in the time of data collection. We understand your concern and would like to address it.

In our study, we acknowledge the potential impact of the variation in data collection time as a limitation in the discussion section. We recognize that there may be temporal differences in factors that could influence the outcomes under investigation. While it is true that the data sets from different countries may have been collected at different times, we have taken several factors into consideration when combining the data sets for analysis. These factors include the similarity in data collection methods, survey instruments, and the relatively close time proximity between the data sets.

Moreover, we have reviewed relevant literature and studies that have successfully utilized similar approaches in combining and analyzing DHS data from multiple countries. These studies have demonstrated the feasibility and validity of combining data sets collected at different times when the contextual factors are similar. In our revised manuscript, we put further emphasize these considerations and provide additional discussion on the potential implications and limitations of combining data sets from different time periods. We ensured that readers have a clear understanding of the steps we have taken and the limitations associated with the time variation in our data sets. Thank you for your valuable feedback.

9. Cluster analysis have to conducted to identify patterns and trends in the data, such as the characteristics and behaviors of different segments of the population; explore the heterogeneity and diversity of the data; evaluate the impact and effectiveness of interventions or programs, such as the differences in outcomes or indicators across clusters; and inform policy and decision-making, such as the allocation of resources or the targeting of interventions to specific clusters.

Response: Dear reviewer, thank you for your feedback regarding the cluster-based analyses and their recommendations to the policy makers and implementor. We have put all the recommendation based on our findings in the conclusions section. Thank you.

10. Did the authors do sensitivity analysis? If not, I recommend the authors do a sensitivity analysis. Because it is essential to testing the robustness and reliability of the results of a data analysis by varying the assumptions, parameters, or data used in the analysis. Additionally Sensitivity analysis can help the authors to assess the impact of uncertainty, variability, or bias on the estimates or conclusions of their analysis.

Response: Dear reviewer, thank you for your suggestion to perform a sensitivity analysis for the multilevel analysis. We appreciate your feedback and would like to elaborate on our rationale for not including a sensitivity, and or a receiver operating characteristic (ROC) analysis in our study. In our research, we focused on comparing the random effects and conducting model comparisons using several measures, including the proportional change in variance (PCV), intraclass correlation coefficient (ICC), median odds ratio (MOR), log-likelihood ratio (LLR), AIC, and deviance information criterion (DIC). These measures are commonly used in multilevel modeling to assess the goodness of fit and compare different models.

We believe that these model comparison metrics provide sufficient information to 

---

## [Decision Letter · Decision Letter 1]

28 Mar 2024

Determinants of Early Initiation of Breastfeeding Following Birth in West Africa: A Multilevel Analysis Using Data from Multi-Country National Health Surveys

PONE-D-23-39493R1

Dear Dr. Terefe,

We’re pleased to inform you that your manuscript has been judged scientifically suitable for publication and will be formally accepted for publication once it meets all outstanding technical requirements.

Kind regards,

Anthony Mwinilanaa Tampah-Naah

Academic Editor

PLOS ONE

Additional Editor Comments (optional):

Reviewers' comments:

Reviewer's Responses to Questions

**Comments to the Author**

1. If the authors have adequately addressed your comments raised in a previous round of review and you feel that this manuscript is now acceptable for publication, you may indicate that here to bypass the “Comments to the Author” section, enter your conflict of interest statement in the “Confidential to Editor” section, and submit your "Accept" recommendation.

Reviewer #1: All comments have been addressed

Reviewer #2: All comments have been addressed

2. Is the manuscript technically sound, and do the data support the conclusions?

Reviewer #1: Yes

Reviewer #2: Yes

3. Has the statistical analysis been performed appropriately and rigorously? 

Reviewer #1: Yes

Reviewer #2: Yes

4. Have the authors made all data underlying the findings in their manuscript fully available?

Reviewer #1: Yes

Reviewer #2: Yes

5. Is the manuscript presented in an intelligible fashion and written in standard English?

Reviewer #1: Yes

Reviewer #2: Yes

6. Review Comments to the Author

Reviewer #1: My comments are sufficiently addressed, the manuscript is scientifically sound, and it is publishable.

Reviewer #2: (No Response)

7. PLOS authors have the option to publish the peer review history of their article (what does this mean?). If published, this will include your full peer review and any attached files.

Reviewer #1: No

Reviewer #2: No

---

## [Editor Report · Acceptance letter]

2 May 2024

PONE-D-23-39493R1 

PLOS ONE

Dear Dr. Terefe, 

I'm pleased to inform you that your manuscript has been deemed suitable for publication in PLOS ONE. Congratulations! Your manuscript is now being handed over to our production team.

Kind regards, 

on behalf of

Dr. Anthony Mwinilanaa Tampah-Naah 

Academic Editor

PLOS ONE